# Targeting Mediator Kinase Cyclin-Dependent Kinases 8/19 Potentiates Chemotherapeutic Responses, Reverses Tumor Growth, and Prolongs Survival from Ovarian Clear Cell Carcinoma

**DOI:** 10.3390/cancers17060941

**Published:** 2025-03-10

**Authors:** Wade C. Barton, Asha Kumari, Zachary T. Mack, Gary P. Schools, Liz Macias Quintero, Alex Seok Choi, Karthik Rangavajhula, Rebecca C. Arend, Eugenia V. Broude, Karthikeyan Mythreye

**Affiliations:** 1Division of Gynecologic Oncology, Department of Obstetrics & Gynecology, Heersink School of Medicine, University of Alabama School of Medicine, Birmingham, AL 35294, USA; wcbarton@uabmc.edu (W.C.B.); rarend@uabmc.edu (R.C.A.); 2Division of Molecular Cellular Pathology, Department of Pathology, Heersink School of Medicine and O’Neal Comprehensive Cancer Center, University of Alabama Birmingham, Birmingham, AL 35294, USA; akumari@uab.edu (A.K.); quintero@uab.edu (L.M.Q.); choisw0702@gmail.com (A.S.C.); 3Department of Drug Discovery and Biomedical Sciences, College of Pharmacy, University of South Carolina, Columbia, SC 29208, USA; zmack@mailbox.sc.edu (Z.T.M.); schoolsg@cop.sc.edu (G.P.S.); kvrangavajhula@gmail.com (K.R.)

**Keywords:** clear cell ovarian carcinoma, cyclin, ovarian cancer

## Abstract

Ovarian clear cell carcinoma (OCCC) is a unique histologic subtype of ovarian cancer that carries a worse prognosis due to its aggressive spread, high rates of platinum resistance, and limited treatment options. In this study, we reveal that CDK8 levels are elevated in OCCC tissues and demonstrate that inhibiting the kinase activity of CDK8/19 with SNX631-6 enhances the efficacy of platinum-based chemotherapy and paclitaxel in both platinum-sensitive and resistant OCCC cell lines. While CDK8/19 inhibition has limited effects on cell proliferation in vitro, SNX631-6 shows strong antitumor and anti-metastatic effects in vivo. Combining SNX631-6 with platinum therapy significantly reduces tumor growth and extends survival, indicating its potential to overcome platinum resistance in OCCC patients. Our data suggest that CDK8/19 inhibition can provide meaningful clinical benefit with limited adverse effects.

## 1. Introduction

Ovarian cancer (OC) is the leading cause of mortality among gynecological malignancies, with the American Cancer Society estimating 20,000 new cases and 13,000 deaths as having occurred in 2024. Despite management with the traditional modalities of debulking surgery and chemotherapy, the 5-year survival rate for OC remains less than 50% [1,2,3]. Additionally, even in those with an initial response to therapy, 75% of those with OC will experience disease relapse or progression. Given the lack of effective treatments for relapsed or progressive disease, the development of alternative treatment approaches is paramount.

Ovarian clear cell carcinoma (OCCC), a type of epithelial ovarian carcinoma, differs from the other histological subtypes in terms of pathogenesis, and molecular, genetic, and clinical features. Although each subtype of OC has unique molecular and clinical features, all epithelial OCs are still treated similarly with upfront surgical debulking in combination with platinum-based chemotherapy. Because OCCC is rare in the United States and European countries, cases have not been actively enrolled in clinical trials, and trials specifically targeting OCCC have been limited. Furthermore, existing molecular studies have been hampered by small sample sizes and a lack of comprehensive testing. Given this, patients with OCCC traditionally have worse stage-for-stage survival outcomes [4] when compared to other histological types, further highlighting the potential need for targeted therapies to fill a much-needed gap.

Together with Cyclin C (CCNC), MED 12, and MED13, cyclin-dependent kinase 8 (CDK8) or its isoform CDK19 form the Mediator kinase module which regulates the core Mediator complex to activate or repress transcription [5,6,7,8,9]. Notably, CDK8/19 influences Pol II phosphorylation selectively, primarily affecting newly induced genes and de novo transcription, while having little impact on basal transcription [10,11]. Given the pivotal role of transcriptional reprogramming during cancer progression and in drug resistance, selective inhibitors targeting RNA polymerase II-associated kinases have gained attention and are currently under clinical evaluation. Specifically, CDK8/19 inhibitors (CDK8/19is) [12] are undergoing clinical trials for both solid tumors and leukemia, as evidenced by entries on www.clinicaltrials.gov (NCT03065010, NCT04021368, NCT05052255, NCT05300438). However, their effectiveness in OC remains unexplored.

In the present study, we explore CDK8/19 levels and the therapeutic value of inhibiting the kinase activity of CDK8/19 in OCCC using SNX631-6, a potent, selective, and non-toxic CDK8/19i in the clinical pipeline [13]. SNX631-6 is 10 times more potent in a cell-based assay [14] than the first CDK8/19i to enter clinical trials (www.clinicaltrials.gov NCT03065010) and lacks the off-target activities that were responsible for the systemic toxicity of two CDK8/19is developed by another group [15,16]. We discovered that CDK8 is upregulated specifically in primary and metastatic human OC, with significant differences in OCCC. We demonstrate for the first time that CDK8/19i strongly potentiates platinum and/or taxane-based chemotherapeutics. In vivo, this effect leads to the inhibition of tumor growth and even tumor regression for established disease. These results suggest the potential utility of combining standard-of-care chemotherapy with CDK8/19is in the treatment of OCCC.

## 2. Materials and Methods

### 2.1. Cell Lines, Culture Conditions, and Reagents

Ovarian epithelial carcinoma cell line SKOV-3 was obtained from ATCC. ES2-Luc cells originated from Dr. Rebecca Arend’s lab. Cell lines TOV21 and RMG-1 were a kind gift from Michael Birrer [17]. All cell line authentication was performed at the Heflin Center for Genomic Science Core Laboratories at The University of Alabama (UAB) using short tandem repeat analysis. SKOV-3, SKOV-3 CDK 8/19 CRISPR dKO, TOV-21-G, and RMG-1 cell lines were grown in RPMI-1640, 10% FBS, and 100 IU/mL penicillin-streptomycin. ES2-Luc cells were maintained in McCoy’s medium, 10% FBS, and 100 IU/mL penicillin-streptomycin. All cell lines were contained at 37 °C in a humified incubator at 5% CO_2_ and routinely checked for mycoplasma contamination using LookOut^®^ Mycoplasma PCR Detection Kit (Sigma-Aldrich, Saint Louis, MO, USA). Experiments were conducted within five to seven passages depending on the cell line. The CDK8/19is SNX631 and SNX631-6 [13,18] were provided by Senex Biotechnology (Columbia, SC, USA).

### 2.2. Cell Proliferation and IC 50 Assays

To determine inhibitory concentration (IC_50_) values for individual lines, cells were seeded into 96-well plates at indicated densities (Table 1) and incubated until 50–75% confluent. After 24 h, cells were treated with carboplatin (0–50 mM), cisplatin (0–50 mM), or paclitaxel (0–100 nM) alone and in combination with SNX631 500 nM. DMSO was used as a vehicle. Cells were left for seven days at 37 °C and 5% CO_2_ in the appropriate medium before cell densities were measured by sulphorhodamine B sodium salt (SRB). Absorbance was measured at 570 nm using a Synergy H1 microplate reader (Biotek, Winooski, VT, USA). Potentiation analysis was determined using GraphPad-Prism based on IC_50_ comparisons between single and combination therapy.

### 2.3. Tissue Specimen Collection

IRB approval was obtained before all human specimen collection. Tissue from patients who had previously undergone surgical debulking for OCCC was identified by the University of Alabama Birmingham’s Tissue Biorepository (TBR) with the generation of formalin-fixed paraffin-embedded tissue microarrays (TMAs) and individual tumor sections.

### 2.4. Immunoblotting

A total of 5 × 10^5^ cells per well were plated in six-well plates and allowed 48 h to attach before being lysed with radioimmunoprecipitation assay buffer containing protease and phosphatase inhibitors. Whole-cell lysates were then resolved on 7.5% to 12% PAGE gels, transferred to Immobilon-P PVDF membranes, blocked with 5% non-fat milk or Immobilon Signal Enhancer, and incubated with primary antibodies overnight at 4 °C: phospho-STAT1 Ser727 (1:1000), CDK 8 (1:1000), Cyclin C (1:1000), MED 12 (1:2000), *ARID1A* (1:1000), b-actin (1:1000), and vinculin (1:1000). Membranes were then washed with TBST and blotted with anti-Mouse or anti-Rabbit IR-dye conjugated secondary antibodies. Protein bands were then detected on an Odyssey CLx. Imager (LicorBio, Lincoln, NE, USA). All uncropped Westerns can be found in Appendix A.

### 2.5. Immunohistochemistry (IHC), H&E Staining, Immunofluorescence (If) Labeling, and Analysis

Formalin-fixed and paraffin-embedded tissue blocks were cut into 10 μM sections, dewaxed, and then rehydrated. Heat-induced epitope retrieval was performed within sodium citrate buffer (pH 6.0). For IHC, endogenous peroxidases were blocked with 3% hydrogen peroxide and tissues were incubated with primary antibody within a humidified chamber at 4 °C overnight (Ki67, (CST: D3B5, IHC Formulated #12202 1:200 and anti-CD31 (1:100, Cat no. D8V9E #77699, CST))) diluted in Da Vinci Green diluent (Biocare Cat no. PD900). Tissues were then treated with an appropriate secondary antibody conjugated to horseradish peroxidase (HRP) in a blocking solution. HRP signal was then detected with 3,3′-diaminobenzidine (DAB) substrate, washed, and then counterstained with Mayer’s hematoxylin.

Stained tumor sections were digitally scanned using the EVOS m7000 microscope (Thermo Fisher, Waltham, MA, USA) system with 20× magnification, with ten areas from each cross-section imaged and converted into .tiff format for compatibility with ImageJ software version 1.54p (NIH, Madison, WI, USA). For quantification of the proliferative index in tumors, Ki67 was quantified within each section manually before being normalized to the total nuclear count. Individual normalized values were pooled together to generate an average proliferative index for each treatment group. CD31+ blood vessels were quantified in non-necrotic regions for each treatment group [19].

For IF of formalin-fixed and paraffin-embedded tissue blocks, tissues were blocked in 5% normal donkey serum in TBST for 1 h. The following primary antibodies were diluted in 1% BSA in TBST and then incubated on the sections overnight at 4 °C: CDK8 (1:400 dilution) and Ki67 (1:2000 dilution). Alexafluor555 and Cy5 labeled secondary antibodies were used at 1:200 for 1 h. Nuclei were counterstained with 1 mM 4′,6-diamidino-2-phenylindole (DAPI) and sections were mounted in Prolong Glass. For IF analysis of tissue microarrays, image stacks were acquired on Agilent (BioTek, VT, US) Cytation 5 with a 4×/0.13 n.a. objective, using identical acquisition parameters for all TMAs and whole sections. Montages of each TMA or section were acquired. Images shown are maximum-intensity projections after the application of a 2-pixel median filter in ImageJ. For fluorescence intensity measurements, tumor regions of interest were manually defined using the DAPI channel to select at least three areas of cancer cells in each section. CDK8 and Ki67 mean pixel intensity quantification was measured within these regions of interest using ImageJ. The labeling, imaging, and quantification were repeated three times, with similar results.

### 2.6. CDK8/19 CRISPR Knock-Out Generation

Gene-specific CRISPR knockout lentiviral constructs (lentiCRISPR-Puro-sgCDK8 and lentiCRISPR-Blast-sgCDK19) were generated by cloning annealed double-stranded oligos with gene-specific sgRNA [14] into the BsmBI site of the lentiviral vector lentiCRISPR v2 or lentiCRISPR v2-Blast. The CDK8/19 double-knockout derivative of SKOV-3 cells was produced through a two-step process involving lentiviral transduction and selection. Initially, SKOV-3-19KO was generated using lentiCRISPR-Blast-sgCDK19, followed by a 5-day incubation with 10 μg/mL blasticidin for selection. Subsequently, SKOV-3-19KO underwent transduction with lentiCRISPR-Puro-sgCDK8, followed by incubation with 5 μg/mL puromycin for selection.

### 2.7. Public Data Mining

DepMap *(*https://depmap.org/portal/, URL initially accessed on 1 June 2023, last accessed 1 December 2024) was used to analyze the expression of CDK8 in a panel of tumor cell lines using the May 2019 updated data. Data were then reformatted into a heat map using GraphPad Prism 9 version 10.4.1 (Dotmatics, San Diego, CA, USA).

### 2.8. Mouse Xenograft Models

All mouse studies were approved by the University of Alabama (UAB) Institutional Animal Care Use Committee and conducted at the UAB Department of Laboratory Animal Research. Female nu/nu mice (aged 6 weeks) were obtained from The Jackson Laboratory and allowed to habituate for 3–5 days while given a regular diet. Mice were then injected either subcutaneously (s.c.) into the right flank with 1 × 10^6^ ES2-Luc cells or directly into the peritoneal cavity with 0.75 × 10^6^ ES2-Luc cells. Once injected, tumors were allowed to establish for 72 h. Mice were then randomized into four treatment groups: vehicle, cisplatin alone, SNX631-6 alone, or cisplatin combined with SNX631-6 [18]. The vehicle group received a control diet from the UAB animal research facility. Cisplatin was administered twice weekly via intraperitoneal (i.p.) injection at a dose of 2 mg/kg diluted in a vehicle of PBS. SNX631-6 was administered in a medicated diet, ad libitum (350 ppm, providing a daily dose of 30–50 mg/kg) due to sustained availability [13]. Tumor volume was recorded twice weekly using calipers, with tumor volume being calculated as [length (mm) × width (mm)^2^]/2. Intraperitoneal (i.p.) burden of disease was measured at least weekly using bioluminescent imaging (see separate section). Weights were recorded at least twice weekly for all mice. For s.c studies, mice were sacrificed per IACUC guidelines when any of the following criteria were met: sustained loss of 20% body weight on two consecutive measurements; maximum tumor volume achieved; tumor ulceration of >10% total tumor volume; or subjective determination of animal distress, as determined by the investigators. For i.p. studies, mice were sacrificed per IACUC guidelines, which include the assessment of body condition scores. Once sacrificed, tumors were excised, weighed, and then either fixed in 10% formalin for 24 h or stored in 70% ethanol at 4 °C.

### 2.9. Animal Bioluminescent Imaging (BLI)

The development of tumors and metastasis was detected using an in vivo imaging system (IVIS (PerkinElmer, Waltham, MA, USA)) three days after i.p. cell injection. Luciferin (150 mg/kg, i.p.) was injected 10 min before imaging was performed. The mice were then anesthetized using isoflurane and imaged dorsally. The region of interest (ROI) was selected and the whole-body luminescence value was measured by Living Image Software v4.8.2 (Revvity, Boston, MA, USA).

### 2.10. Anesthesia

Mice were anesthetized using pharmaceutical-grade isoflurane dispensed from a precision vaporizer in 100% oxygen delivery gas at 2% to an induction chamber. For IVIS imaging (PerkinElmer, Waltham, MA, USA), there was a dedicated XGI8 gas inhalation anesthesia apparatus next to the imager. Once inside the IVIS imager, anesthesia was continued with isoflurane at 2% via a face mask. After the imaging was performed, mice were monitored until they were able to ambulate normally before returning to their assigned husbandry rooms.

### 2.11. Statistical Analyses

All data are representative of three independent experiments unless otherwise described in legends. Statistical analysis was performed using GraphPad Prism 9, with statistical tests chosen based on the experimental setup and specifically described in the figure legends. Data are expressed as mean ± SEM. The difference between the two groups was assessed using a nonpaired *t*-test. Multiple group comparisons were carried out by the analysis of variance (ANOVA) using one-way ANOVA followed by appropriate post hoc tests, as indicated in figure legends. Survival and time-to-event curves were analyzed using log-rank statistics. The Mann–Whitney test was performed to test for the CDK8 and Ki67 distribution change in tissue arrays. A *p*-value of≤0.05 was considered significant for all analyses.

## 3. Results

### 3.1. CDK8/19 Is Significantly Expressed in Both Primary and Metastatic Clear Cell Ovarian Carcinoma and Correlates with Poor Prognosis

To investigate CDK8 protein levels in situ, in a panel of unmatched malignant and nonmalignant tissues from different OC subtypes, immunofluorescence (IF) was performed (Figure 1(Ai)). Quantitation of the intensity of CDK8 across duplicate cores from the different ovarian cancer subtypes revealed higher levels of CDK8 in all tumor tissues as compared to benign tissue that comprised a combination of ovary, omentum, and fallopian tube tissues (Figure 1(Ai,Aii)). The most statistically significant differences were observed in mucinous and clear cell subtypes (Figure 1(Aii)). Notably, metastatic tissue from clear cell tumors were further elevated for CDK8 levels compared to both benign and primary cores (Figure 1(Aii)). We thus examined OCCC tissue for Ki67 staining to identify proliferative regions and assess the distribution of CDK8 and Ki67 (Figure 1(Bi)). While both primary and metastatic tumors exhibited a full range of CDK8 distribution, the CDK8 and Ki67 (Figure 1(Bii)) immunoreactivities in the metastatic tissues were concentrated qualitatively in the tumoral and cystic regions. We further quantified the colocalization of CDK8 and Ki67 in the OCCC primary and metastatic cores. Pearson and Manders’ correlation between CDK8 and Ki67 trended to be higher in metastasis samples (Figure 1(Bii)). These data together indicate that CDK8 protein levels are elevated in OC tissues, particularly in mucinous and clear cell subtypes. Additionally, metastatic clear cell tumors appear to demonstrate higher levels of CDK8 compared to primary or benign tissues.

Based on the significant differences in overall CDK8 levels in OCCC tissues, we investigated the levels of CDK8 and the Mediator-associated CDK module components CCNC and MED12, in OCCC cell lines with known *ARID1A* mutations (TOV-21-G) and in those with no *ARID1A* mutations (ES-2, RMG-1). *ARID1A* status stratification was chosen due to the high incidence of *ARID1A* mutations in OCCC and the potential link with intrinsic platinum resistance [20,21]. We also included one non-OCCC cell line (SKOV-3) with previously reported ARID1A alterations [22,23,24]. CDK8, ARID1A, CCNC, and MED12 were detected in all cell lines at the protein level (Figure 1C). Across the cell lines, differences in CDK8, CCNC, and MED12 protein levels were found to be minimal. ARID1A levels mirrored known mutational status with higher levels in the *ARID1A* wild-type (WT) lines ES-2 and RMG-1 and the lowest levels in the *ARID1A*-mutated lines TOV-21-G and SKOV-3. Analysis of publicly available data on the cell lines (depmap.org) indicated that RNA levels for CDK8 were concordant with protein levels in the cell lines (Figure 1D). Among the evaluated cell lines, ES-2 and TOV-21-G were chosen to model *ARID1A* WT and *ARID1A* mutated models for subsequent experiments.

### 3.2. Targeting CDK8/19 Kinase Activity Is Not Independently Cytotoxic to Ovarian Cancer Cell Lines

Independent inhibition of CDK8/19’s kinase activity does not significantly impact steady-state growth in vitro in most tested cancers except for some leukemia [25], and moderate responses in some breast [26], colon [27], and prostate cancers [28,29]. Therefore, to determine the sensitivity to the kinase inhibition of CDK8/19 in OCCC models, we utilized SNX631, a specific inhibitor of CDK8/19 (CDK8/19i) [14,18].

Selected cell lines were divided into *ARID1A* WT (ES-2, RMG-1) and *ARID1A* mutations (TOV-21-G). SKOV-3 was included as a mixed-histology (clear cell, serous) and ARID1A-altered model [22]. Monotherapy treatment with DMSO (vehicle), cisplatin (0–50 µM), carboplatin (0–50 µM), paclitaxel (0–100 nM), and SNX631 (0–3 µM) was conducted for seven continuous days (Figure 2A–D). Cells exposed to cisplatin treatment produced IC_50_ values ranging from 4.21 µM for ES-2 to 29.6 µM for SKOV-3, replicating SKOV-3’s known primary platinum resistance. Similarly, single-agent carboplatin IC_50_ values ranged from 12.20 µM for ES-2 to 210.31 µM for SKOV-3. All tested cell lines exhibited some intrinsic level of carboplatin resistance, which has not been previously documented. Paclitaxel IC_50_ values from 3.43 nM for SKOV-3 to 15.24 nM for TOV-21-G. CDK8/19 inhibition with SNX631 as a single agent had only minor effects on cellular proliferation over seven days, producing IC_50_ values from 1.41 µM for TOV-21-G to 6.14 µM for SKOV-3. For the chemotherapeutics, any IC_50_ value >/= 10 was correlated with primary resistance [30,31]. To ensure the specificity of the observed responses to the CDK8/19i by SNX631, we generated a CDK8/19 CRISPR dKO model (Figure 2(Ei,Eii)). SNX631 failed to elicit any effect in the SKOV-3 CRISPR dKO cells, further validating the specificity.

### 3.3. CDK8 Inhibition Potentiates Cytotoxicity Towards Traditional Chemotherapeutics

We then tested the pharmacokinetic interaction between cisplatin, carboplatin, paclitaxel, and the CDK8/19i SNX631. The five OC cell lines above were treated for seven continuous days with DMSO (vehicle), and a dose range of cisplatin (0–50 µM), carboplatin (0–50 µM), or paclitaxel (0–100 nM), along with a fixed- sub-therapeutic dose of SNX631 (500 nM). A dose of 500 nM of SNX631 was previously shown to be sufficient for complete CDK8/19 kinase activity inhibition [18] and reduced the phosphorylation of STAT1 at S727, which is phosphorylated directly (but not exclusively) by CDK8/19 [16,32] (Appendix A), without significant cellular growth inhibition. Moreover, as previously demonstrated [18], SNX631 shows IC50 values of 7–11 nM in different cell-based assays for its transcription co-factor activity, and a 500 nM concentration leads to the maximal transcriptional effect. Here, in this study, we found that this concentration did not inhibit ovarian cancer cell growth (Figure 2). Therefore, instead of synergy analysis, potentiation analysis was tested as the endpoint, defined as a left shift in the fixed-ratio IC_50_ curve relative to the monotherapy IC_50_ curve.

All cell lines, independent of histology, demonstrated some degree of IC_50_ left shift, indicating growth inhibition potentiation with SNX631 combination therapy (Figure 3A–D). Notably, platinum-resistant, *ARID1A*-mutated TOV-21-G exhibited significant IC_50_ left shifts when exposed to the combination of platinum chemotherapy and SNX631 (Figure 3B). Additionally, despite our categorization of TOV-21-G as an intrinsically paclitaxel-resistant model based on a monotherapy IC_50_ value of 10.52 nM, the combination of paclitaxel with SNX631 reversed this response, with the re-sensitization of cells to taxol treatment (Figure 3H). In the case of ES-2 and RMG-1 (platinum-sensitive, *ARID1A* WT), both demonstrated a significant increase in platinum-induced growth inhibition when used concurrently with SNX631 (Figure 3F,G). Significant growth inhibition was also observed in SKOV-3 cells when SNX631 was combined with paclitaxel. In the SKOV-3 CDK8/19 dKO cells, the IC_50_ values for the platinum compounds cisplatin and carboplatin (3.71 µM and 20.73 µM, respectively) were an order of magnitude lower compared to the CDK8/19-intact SKOV-3 (Figure 3E), demonstrating that CDK8/19 inhibition abrogates innate platinum resistance and directly attenuates taxol resistance. Interestingly, unlike in the OCCC lines, a dampened effect on in vitro growth inhibition was seen in SKOV-3 when a CDK8/19i was combined with platinum-based chemotherapy. IC_50_ values for drug combinations in all tested cell lines are shown in Appendix A, with the fold-change relative to single-agent therapy. Hence, combining CDK8/19is with existing chemotherapy in OCCC not only results in the potentiation of drug-induced cytotoxicity but overcomes intrinsic platinum and taxol resistance.

### 3.4. CDK8/19 Inhibition Suppresses In Vivo Tumor Growth and Prolongs Overall Survival from Ovarian Clear Cell Carcinoma

We next tested the effects of in vivo treatment with cisplatin and SNX631-6, an equipotent analog of SNX631 in the clinical pipeline [13] in ES-2 *Foxn1^nu^* xenografts. Both subcutaneous (s.c.) and intraperitoneal (i.p.) injection models were utilized. To ensure tumor establishment, mice were observed 72 h after tumor cell injection with ES-2 parental luciferase-expressing cells before being randomized to treatment with vehicle, cisplatin, SNX631-6, or combination therapy with SNX631-6 and cisplatin. Effects of treatment on tumor volumes, time-to-event, and mouse body weight for s.c. ES-2 xenografts are shown in Figure 4A,B. s.c. tumors in all groups were harvested upon reaching a maximum volume of 3000 mm^3^, which served as the surrogate marker for survival. We found significant tumor growth inhibition (Figure 4A) and a significant increase in the overall survival of mice in the cisplatin and SNX631-6 combination group (Figure 4B), as compared to vehicle (28 days v. undefined), cisplatin monotherapy (30.5 days v. undefined), or SNX631-6 monotherapy (54 days v. undefined). A reduction in overall tumor burden (Figure 4C) and increased overall survival of mice from tumor burden were also observed in the i.p. model (Figure 4D), with a significant survival benefit in the combination therapy compared to vehicle (33 days v. 47 days), cisplatin monotherapy (37 days v. 47 days), or SNX631-6 monotherapy (32 days v. 47 days). Tumor growth inhibition and overall survival benefit were more pronounced in the s.c. (Figure 4B) model compared to the i.p. model (Figure 4D). We suspect this difference to be secondary to a slightly blunted effect on i.p. tumor growth inhibition following significant ascites accumulation due to the aggressive nature of ES2 cells, a phenomenon that occurred in all i.p. treatment groups around week 4. Once ascites developed, mice rapidly became moribund, meeting established criteria for sacrifice. Strikingly, SNX631-6 treatment alone as a single agent also significantly decreased tumor size and tumor burden in both models, despite its weak in vitro effect (Appendix A). There was no treatment toxicity based on decreasing mouse body weight in the combination groups (Appendix A).

**Figure 3 cancers-17-00941-f003:**
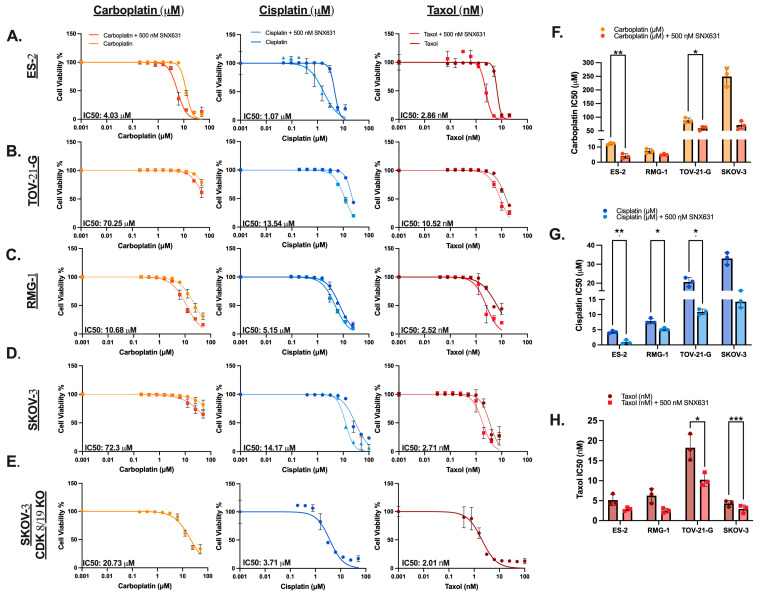
Combining standard-of-care chemotherapy with CDK8/19 inhibition enhances chemotherapeutic response in ovarian cancer cell lines. (**A**–**D**) Seven-day dose–response curves for indicated OC cells treated with increasing concentrations of carboplatin (0–50 µM), cisplatin (0–50 µM), or taxol (0–100 nM) alone or in combination with 500 nM SNX631. Cell viability data are expressed as percentage SRB measurements relative to vehicle-treated cells. IC_50_ values represent n = 6 experiments with six replicates per drug concentration. Combination IC_50_ values are shown in the lower left corner. (**E**) Seven-day dose–response curves for SKOV-3 CDK8/19 dKO cells combined with indicated chemotherapeutics. (**F**–**H**) A quantitation of the IC_50_ change between (**F**) carboplatin, (**G**) cisplatin, or (**H**) taxol monotherapy and combination therapy across indicated OC lines. (Mean SEM (n = 6)). *, *p* < 0.05; **, *p* < 0.01; ***, *p* < 0.001, unpaired *t*-test.

The assessment of harvested tumors from both s.c. and i.p. groups (Appendix A) from all treatment arms demonstrated tumor necrosis (necrosis indicated by black arrow, Appendix A), including in the control group, which agrees with the previous characterization of this xenograft model [33]. Both individual drugs and their combination strongly decreased staining for the proliferation marker Ki67 in the s.c. (Figure 4E,F) and i.p. tumor models (Figure 4G,H). Significance was reached between control v. cisplatin (*p* < 0.0001) and control v. combination treatment groups (*p* < 0.0001).

In addition to proliferation differences, gross tumor morphology assessments (Appendix A) appeared less vascular. CD31 analysis of the blood vessels in the s.c tumors revealed a significant reduction in the number of blood vessels, particularly in those treated with the combination of SNX631 and Cisplatin (*p* < 0.0001), as compared to the control group (Appendix A). Taken together, these data confirm the enhanced efficacy of CDK8/19is, particularly when combined with platinum-based chemotherapy, evidenced by the inhibition of tumor growth, decreased cellular proliferation and vascularity, and improved overall survival.

## 4. Discussion

OC is largely incurable, with a poor prognosis independent of tumor molecular heterogeneity. Despite aggressive treatment with upfront surgical debulking and platinum-based chemotherapy, recurrent disease develops in more than 80% of women, with a 10-year disease-free survival rate below 15%. Therefore, the development of novel molecularly targeted therapies represents an urgent, unmet clinical need. CDK8 has been outlined as a proto-oncogene, with multiple studies correlating elevated CDK8 expression to poor clinical prognosis in various solid malignant tumors [34], including ovarian cancer [35]. In this study, we explored the effect of CDK8/19 inhibition on OC tumor growth, disease progression, and chemotherapeutic response, showing for the first time that the use of a CDK8/19i, especially in combination with platinum-based therapy, is clinically translatable for gynecological cancers.

Here, we observed that CDK8 was elevated in all OC, with primary and metastatic OCCC tumor tissues expressing the highest levels of CDK8 compared to non-malignant/benign tissue samples. The metastatic tissues trended towards enhanced CDK8 expression in other subtypes as well. Notably, elevated expression of Mediator kinases in tumors is not required for the inhibitors to selectively impact tumor cells. This is attributed to the findings that show a CDK8/19i selectively impacts phenotypes driven by transcriptional reprogramming, and the contextual functions of CDK8/19 as both a positive and negative regulator of stress and oncogene-induced transcription [10,18,36].

OCCC presents distinct molecular features compared to other EOC subtypes, with ARID1A mutations occurring in approximately 50% of cases [37,38]. The effective rate of platinum sensitivity in OCCC is less than 50% and primary resistance is commonly seen [39,40]. Our studies suggest that both ARID1A wild-type and mutant cell lines showed similar responses to CDK8/19 inhibition, suggesting that response to CDK8/19is may be independent of ARID1A status. This finding is particularly relevant as OCCC patients currently receive standard platinum-doublet therapy despite primary resistance rates [40,41], highlighting the need for novel therapeutic approaches. Our in vivo studies reveal strong anti-tumor activity of SNX631-6 both as monotherapy and in combination with cisplatin, with strikingly limited cytotoxicity in vitro. Cisplatin was chosen as the platinum-containing compound due to its historic utilization in gynecological cancers. Cisplatin monotherapy efficacy could be inferior to newer compounds such as carboplatin, which are challenging to use in mouse models [41].

Some early CDK8/19is were hampered by reports of toxicity; however, such toxicity was later shown to be due to off-target binding of prior compounds [16]. While CDK8 is required for embryonic development due to the significance of transcriptional reprogramming at that stage, tamoxifen-induced depletion in adult mice has no overt phenotype [42,43]. Additionally, most human cell lines screened in the Broad Institute’s Dependency Map project do not depend on CDK8, suggesting that inhibition-related collateral toxicity would be limited. In the first Phase I clinical trial of a CDK8/19i, Senexin B (NCT03065010), the compound was well-tolerated in advanced breast cancer patients for up to eight months without adverse drug effects. Additional CDK8/19is have successfully entered further clinical trials (NCT03065010, NCT04021368, NCT05052255, NCT05300438).

## 5. Conclusions

While single-agent CDK8/19is were not significantly cytotoxic in vitro, combination of a sub-cytotoxic dose with chemotherapies was highly effective in increasing the cytotoxicity of the chemotherapy. CDK8/19is have also been shown to be broadly effective in potentiating conventional and targeted drugs, radiation, and immunotherapies [18,26,35,44,45,46,47,48]. Additionally, single-agent CDK8/19i treatment led to significant tumor growth inhibition in vivo without any toxicity. Selective CDK8/19i treatment is well tolerated in exceptionally long-term in vivo studies (up to 300 days) [13]. The enhanced in vivo efficacy may be attributable to the effects of CDK8/19i on the tumor microenvironment (TME) caused by suppressing tumor-promoting paracrine activities of stromal fibroblasts and stimulating the antitumor activity of NK cells [35,45], as seen in other models. Mediator kinases have also been implicated in angiogenesis [49] and, notably, tumors in our treatment groups were qualitatively less vascular (Appendix A).

The present study reveals that targeting CDK8/19i can provide a strategy for potentiating the cytotoxic effects of existing chemotherapeutics used in OCCC treatment and proposes potent CDK8/19is as a viable option for further exploration and optimization in the treatment of ovarian cancers.

## Figures and Tables

**Figure 1 cancers-17-00941-f001:**
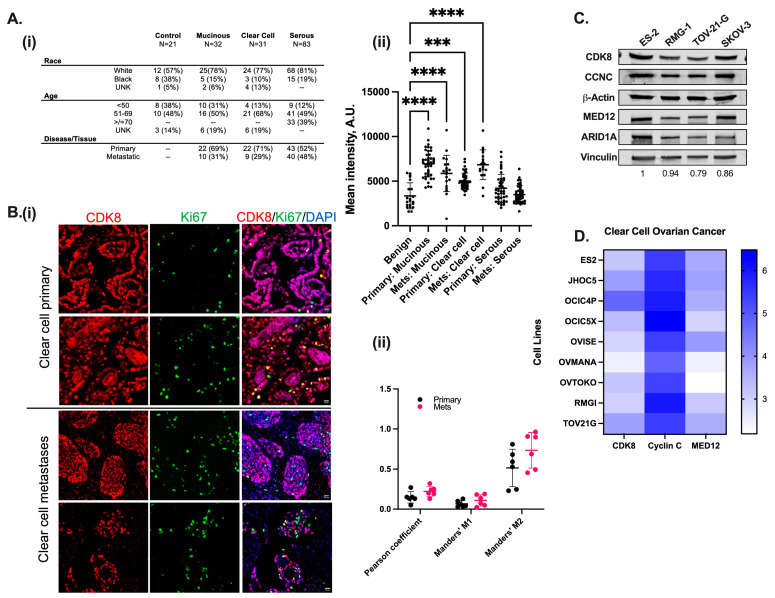
CDK8/19 protein expression is highest in both metastatic and primary clear cell ovarian carcinoma. (**A**) (i) Sample information on the tissue microarrays. (ii) The mean fluorescence intensity across all cores stained for CDK8, with each patient analyzed in duplicate cores. (**B**) (i) Representative images of CDK8 and Ki67 showing the distribution pattern in a subset of cores from the same tissue arrays as in (**A**). Images from each section were adjusted to optimally show the labeling pattern, independent of signal intensity. (ii) The colocalization of CDK8 and Ki67 in six representative OCCC sections. Pearson’s coefficient displays the overall correlation between the two markers and Manders’ M2 displays the fraction of Ki67+ areas that are also CDK8+. (**C**) Western blots from listed cell lines assessing the baseline expression of CDK8, MED12, Cyclin C (CCNC), and *ARID1A*. Quantitative differences in ARID1A normalized to vinculin are included. (**D**) Heatmap comparing CDK8, CCNC, and MED12 expression from available public data (www.Depmap.com) for ovarian clear cell carcinoma cell models. (Mean SEM, ***, *p* < 0.001, ****, *p* < 0.0001). Original western blots are presented in Appendix A.

**Figure 2 cancers-17-00941-f002:**
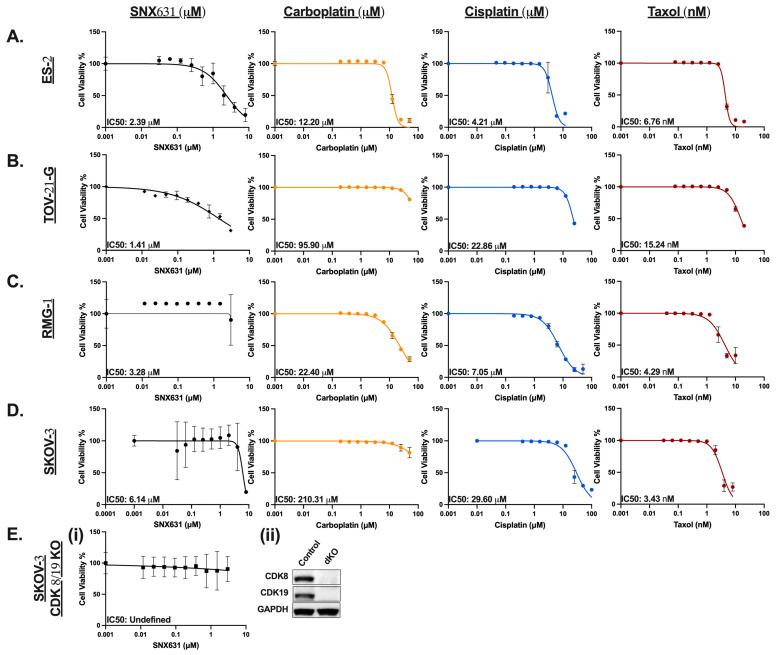
CDK8/19 inhibition and/or loss is not independently cytotoxic to ovarian cancer cell lines. (**A**–**D**) Seven-day dose–response curves for indicated OC cells treated with increasing concentrations of SNX631 (0–3 µM), carboplatin (0–50 µM), cisplatin (0–50 µM), or taxol (0–100 nM) alone. Cell viability data are expressed as percentage SRB measurements relative to vehicle-treated cells. Inhibitory concentration 50 values (IC_50_) represent n = 6 experiments, with six replicates per drug concentration, and are shown in the lower left corner. (**E**) (i) Seven-day dose–response curve in SKOV-3 CDK8/19 dKO cells treated with increasing concentrations of SNX631. (ii) Western blot of CDK8 and CDK19 in SKOV-3 CRISPR dKO cells. Original western blots are presented in Appendix A.

**Figure 4 cancers-17-00941-f004:**
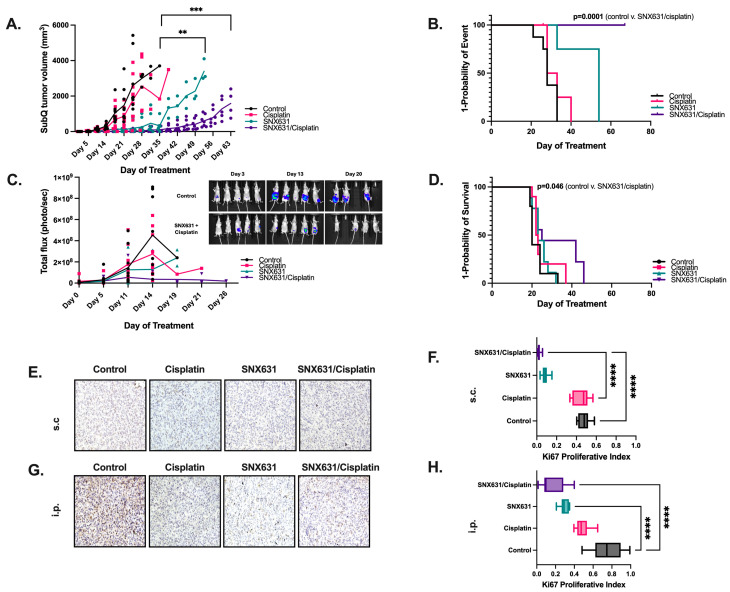
CDK8/19 inhibition monotherapy and in combination with chemotherapy inhibited in vivo tumor growth and metastasis and prolonged survival. (**A**) Subcutaneous (s.c.) tumor volume changes for ES2-luc NU/J xenografts treated with control (PBS, n = 9), cisplatin 2mg/kg twice weekly (n = 10), oral SNX631-6 (n = 10), or cisplatin 2 mg/kg twice weekly + SNX631-6 combination (n = 10) with (**B**) time-to-event of end-point tumor volume. (**C**) Luminescence in i.p. cavity over time of intraperitoneal tumor growth of ES2-luc cells in NU/J mice treated as described (inset: representative luminescence images of mice at indicated days before significant fluid/ascites accumulation). (**D**) Time-to-event as defined by survival endpoints for injected mice in respective treatment groups. (**E**–**H**) Representative images of Ki67 labeling in subcutaneous (**E**,**F**) and (**G**,**H**) intraperitoneal ovarian tumors. Proliferative index normalized to total nuclei from 10 random fields calculated for N = 3 mice per group and presented in (**F**,**H**). (Mean SEM, (n = 3); **, *p* < 0.01; ***, *p* < 0.001, ****, *p* < 0.0001; one-way ANOVA followed by Tukey’s multiple comparison).

**Table 1 cancers-17-00941-t001:** Respective cell lines and plating density to determine IC_50_.

Cell Line	Initial Cell Density (Cell/Well)
ES-2 Luc	2000
TOV-21-G	2000
RMG-1	2500
SKOV-3	1500
SKOV-3 CDK 8/19 KO	1500

## Data Availability

All original Western blots are provided in the Appendix A. Additional images are available upon request. The study does not report any original code.

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
