# Peer review of "Targeting Mediator Kinase Cyclin-Dependent Kinases 8/19 Potentiates Chemotherapeutic Responses, Reverses Tumor Growth, and Prolongs Survival from Ovarian Clear Cell Carcinoma"

_cancers, 2025, doi:10.3390/cancers17060941_

Round 1
Reviewer 1 Report
Comments and Suggestions for Authors
Overall, this work is of interest to the GYO field given clear cell carcinoma is often extremely platinum resistant and this mechanistic approach to overcome chemoresistance in general could potentially alleviate the morbidity/mortality of this disease type.
Overall, the data is reasonable but there are some concerns/recommendations:
1. Using a KO of CDK8/19 as a control with clear cell (rather than SKOV-3 which is typically viewed as a serous histology) would have made more mechanistic sense. Esp in ARID1A WT and ARID1A mut lines. Additionally, determining if these cell lines themselves with KO have less capability of initiation in vivo and metastatic spread also is important and missing given the proposed mechanistic pathway
2. Recommend for figure 3 to determine the true combinatorial index for these drugs. It seems that the dosing was arbitrary and therefore additional true CI testing should be completed or "potentiation" cannot definitely be proven (i.e the shifting of the IC50's while perhaps statistically significant is minimal based on the curves). In fact, what we find with these types of studies is often
3. I am concerned about the definition of "meaningful" in the in vivo experiments for ip. These are the most important experiments and seeing only a 2-3 day diffrerence in survival between the groups is concerning. The authors note this is due to ES-2 growth and ascites, but there ARE other models and one would like to see an ip model benefit (given this is indeed where this tumor grows!) rather than just an sc benefit. Would recommend repeating either in PDX models of clear cell with known genetic makeup or attempting to use a different cell line as this would be more accurate depiction of a "true" tumor rather than a cell line based assay with limited efficacy. Additionally, please see #1 above.
4. Of note, in supp figures, the tumors truly dont seem "smaller" however they do appear more fibrotic and perhaps less vascular. Would recommend considering looking at vessel density in these tumors as well
Author Response
Reviewer 1:
Overall, this work is of interest to the GYO field given clear cell carcinoma is often extremely platinum resistant and this mechanistic approach to overcome chemoresistance in general could potentially alleviate the morbidity/mortality of this disease type.
Overall, the data is reasonable but there are some concerns/recommendations:
- Using a KO of CDK8/19 as a control with clear cell (rather than SKOV-3 which is typically viewed as a serous histology) would have made more mechanistic sense. Esp in ARID1A WT andARID1A mut lines. Additionally, determining if these cell lines themselves with KO have less capability of initiation in vivo and metastatic spread also is important and missing given the proposed mechanistic pathway
We fully agree with the reviewer that SKOV-3 is not a clear cell line which we have stated transparently in the text. SKOV-3 are also not considered high grade serous. However, this cell line exhibits ARID1A and PIK3Ca mutations like clear cells. Our sole purpose of conducting the KO of CDK8 and CDK19 was to demonstrate the selectivity of the pharmacological kinase inhibitor that we used, by determining whether the inhibitor affects cells lacking CDK8/19 and whether the observed effect of the inhibitor (sensitization to cisplatin) can be phenocopied by the KO. Hence, these goals, which we accomplished using SKOV-3 KO (optimal knockout obtained), address the selectivity of the inhibitor and not the biology of clear cell carcinoma, which we indicate is likely independent of the ARID1A status. Tumor initiation analysis was not a focus of the studies, and we share the reviewer’s interest in this question, but anticipate conducting such studies in the future.
Recommend for figure 3 to determine the true combinatorial index for these drugs. It seems that the dosing was arbitrary and therefore additional true CI testing should be completed or "potentiation" cannot definitely be proven (i.e the shifting of the IC50's while perhaps statistically significant is minimal based on the curves). In fact, what we find with these types of studies is often
As previously demonstrated (Ding et al., DOI: 10.1073/pnas.2201073119), the CDK8/19 inhibitor SNX631, used in the present study, shows IC50 values of 7-11 nM in different cell-based assays for its transcription co-factor activity, and 500 nM concentration leads to the maximal transcriptional effect. In the present study, we found that this concentration did not inhibit ovarian cancer cell growth, as the concentrations required for IC50 calculation based on the inhibition of ovarian cancer cell proliferation are in the higher μM ranges, outside the selective range of the drug. Therefore, our analysis combined a fixed 500 nM concentration of SNX631, sufficient for complete CDK8/19 kinase inhibition, with a full concentration range of cytotoxic drugs. This approach is not compatible with traditional combination index (CI) calculations, which are used to determine synergy between two drugs each of which is growth-inhibitory within its meaningful concentration range. Therefore, rather than relying on CI calculations, we assessed the IC50 shift as a more accurate measure of how CDK8/19 kinase inhibition enhances sensitivity to standard-of-care (SOC) treatments. Combining SOC drugs with SNX631 reduced the IC50 by as much as 74% (also see Supp Fig. 2).
I am concerned about the definition of "meaningful" in the in vivo experiments for ip. These are the most important experiments and seeing only a 2-3 day difference in survival between the groups is concerning. The authors note this is due to ES-2 growth and ascites, but there ARE other models and one would like to see an ip model benefit (given this is indeed where this tumor grows!) rather than just an sc benefit. Would recommend repeating either in PDX models of clear cell with known genetic makeup or attempting to use a different cell line as this would be more accurate depiction of a "true" tumor rather than a cell line based assay with limited efficacy. Additionally, please see #1 above.
We regret the typo in the text. As indicated in Fig4D, the survival benefit in the SNX631 mono therapy and combination therapy in the i.p. model was more than 12 days and was statistically significant (p=0.046). We have corrected the language in the revised version of the manuscript. We suspect that the reason for the weaker effect in the i.p. setting is the likely highly aggressive nature of the ES2 i.p. model, which is typical for ovarian clear cell carcinomas. We appreciate the suggestion to do a PDX model, however PDX models for this rare subtype of OCCC are difficult to establish (limited patient population), which we anticipate developing in the long term for future studies.
Of note, in supp figures, the tumors truly don’t seem "smaller" however they do appear more fibrotic and perhaps less vascular. Would recommend considering looking at vessel density in these tumors as well
We thank the reviewer for this comment and helpful suggestion. We have now assessed vascularity in tumors by staining for CD31 and quantifying the number of blood vessels in each group and have indeed found significant differences in vessel number in the combination group. These data are now included in new Supp Fig 7.

Reviewer 2 Report
Comments and Suggestions for Authors
Barton et al. submitted their manuscript entitled “Targeting Mediator Kinase Cyclin-Dependent Kinases 8/19 potentiates chemotherapeutic responses, reverses tumor growth, and prolongs survival from ovarian clear cell carcinoma”.
The Authors of this study present results of a combination therapy of a CDK8/19 inhibitor with platinum or taxan anticancer agents in ovarian clear cell carcinoma (OCCC) cell lines in vitro and in vivo. The research hypothesis is very interesting, inspiring and up-to-date, since OCCC has poor prognosis and there is still no effective treatment for this disease. Design of the experimental study plan is complex including cell-based assays, investigation of human tissue samples and in vivo animal studies. Based on the results a comprehensive overview about the effectiveness of the applied combination therapy has been summarized.
The manuscript has been written in good quality; publication in journal Cancers can be recommended after minor revision of the manuscript.
Minor comments and questions:
a; Line 101: Please correct the concentration of carboplatin and cisplatin “mM” to “µM”. Moreover, the units of measure of concentration or IC50 values in the text of the manuscript and in figure legends are written in incorrect format (most frequently “M” can be found; e.g. Line 278-286 or 294, 303, 324, etc.). Please, check and correct them.
b; Line 105 and 148: Please, add the location of the company Biotek.
c; Line 125: Please, add the name and location of the manufacturer for Odyssey CLx imager.
d; Line 136: Please, add the name and location of the manufacturer for EVOS m7000 microscope.
e; Line 138: Please, add the name and location of the manufacturer for ImageJ software.
f; Line 168 and 209: Please, add the name and location of the manufacturer for GraphPad Prism9 software.
g; Line 180: Why did you administer SNX631-6 in a medicated diet, ad libitum? How did you determine the range of the consumed daily dose of SNX631-6? Did you perform kinetic experiments with SNX631-6 earlier to calculate the consumed daily dose? If yes, please, add as a reference to the text of the manuscript or as supplementary data.
h; Line 198: Please, add the name and location of the manufacturer for Living Image Software.
i; Line 204: Please, add the name and location of the manufacturer for IVIS imager.
j; Line 304: There is a confusing sentence regarding the combination of SNX631 and anticancer agents: “…or paclitaxel (0-100 nM) in a fixed-ratio combination with a sub-therapeutic dose of SNX631 for all cells (500 nM).” If two drugs are combined in a fixed ratio, concentration of both compounds varies in the range of dilution, however the ratio of the two concentrations is constant. Based on the aforementioned sentence, in your experiment, concentrations of the chemotherapeutics changed in a well-defined concentration range, but the concentration of SNX631 was constant (500 nM). Please, rephrase and correct this sentence.
k; Line 317: Is this the correct IC50 value (10.52 nM) of paclitaxel monotherapy in TOV-21-G cell line?
l; Figure 3, panel F-H: Application of segmented axes would be useful to better see the difference between the IC50 values. Please, re-edit the Y-axis of the graphs.
After performing the abovementioned corrections in the text and answering the questions, the manuscript will be suitable for publication in the journal Cancers.
Author Response
Reviewer 2:
Barton et al. submitted their manuscript entitled “Targeting Mediator Kinase Cyclin-Dependent Kinases 8/19 potentiates chemotherapeutic responses, reverses tumor growth, and prolongs survival from ovarian clear cell carcinoma”.
The Authors of this study present results of a combination therapy of a CDK8/19 inhibitor with platinum or taxan anticancer agents in ovarian clear cell carcinoma (OCCC) cell lines in vitro and in vivo. The research hypothesis is very interesting, inspiring and up-to-date, since OCCC has poor prognosis and there is still no effective treatment for this disease. Design of the experimental study plan is complex including cell-based assays, investigation of human tissue samples and in vivo animal studies. Based on the results a comprehensive overview about the effectiveness of the applied combination therapy has been summarized.
The manuscript has been written in good quality; publication in journal Cancers can be recommended after minor revision of the manuscript.
Minor comments and questions:
a; Line 101: Please correct the concentration of carboplatin and cisplatin “mM” to “µM”. Moreover, the units of measure of concentration or IC50 values in the text of the manuscript and in figure legends are written in incorrect format (most frequently “M” can be found; e.g. Line 278-286 or 294, 303, 324, etc.). Please, check and correct them.
We thank the reviewer and have made the corrections.
b; Line 105 and 148: Please, add the location of the company Biotek.
We have now added it.
c; Line 125: Please, add the name and location of the manufacturer for Odyssey CLx imager.
We have now added it.
d; Line 136: Please, add the name and location of the manufacturer for EVOS m7000 microscope.
We have now added it.
e; Line 138: Please, add the name and location of the manufacturer for ImageJ software.
We have now added it.
f; Line 168 and 209: Please, add the name and location of the manufacturer for GraphPad Prism9 software.
We have now added it.
g; Line 180: Why did you administer SNX631-6 in a medicated diet, ad libitum? How did you determine the range of the consumed daily dose of SNX631-6? Did you perform kinetic experiments with SNX631-6 earlier to calculate the consumed daily dose? If yes, please, add as a reference to the text of the manuscript or as supplementary data.
Medicated diet was used for sustained delivery of SNX631-6, due to its limited half-life, and has been described in. (DOI: 10.1172/JCI176709). This has now been added to the manuscript in the Methods.
h; Line 198: Please, add the name and location of the manufacturer for Living Image Software.
We have now added it.
i; Line 204: Please, add the name and location of the manufacturer for IVIS imager.
We have now added it.
j; Line 304: There is a confusing sentence regarding the combination of SNX631 and anticancer agents: “…or paclitaxel (0-100 nM) in a fixed-ratio combination with a sub-therapeutic dose of SNX631 for all cells (500 nM).” If two drugs are combined in a fixed ratio, concentration of both compounds varies in the range of dilution, however the ratio of the two concentrations is constant. Based on the aforementioned sentence, in your experiment, concentrations of the chemotherapeutics changed in a well-defined concentration range, but the concentration of SNX631 was constant (500 nM). Please, rephrase and correct this sentence.
We have now rephrased this sentence to explain the use of a fixed dose.
k; Line 317: Is this the correct IC50 value (10.52 nM) of paclitaxel monotherapy in TOV-21-G cell line?
Yes, as determined for a 7-day period assay as described.
l; Figure 3, panel F-H: Application of segmented axes would be useful to better see the difference between the IC50 values. Please, re-edit the Y-axis of the graphs.
We thank the reviewer and have made the corrections.
After performing the abovementioned corrections in the text and answering the questions, the manuscript will be suitable for publication in the journal Cancer
